# Elemental Compositions of Wood Plastic Pellets Made from Sawdust and Refuse-Derived Fuel (RDF) Waste

Aujchariya Chotikhun [1,*], Rattikal Laosena [2], Jitralada Kittijaruwattana [1], Seng Hua Lee [3], Kanokorn Sae-Ueng [1], Charoen Nakason [1], Yutthapong Pianroj [1] and Emilia-Adela Salca [4]

1 Faculty of Science and Industrial Technology, Prince of Songkla University, Surat Thani Campus, Mueang Surat Thani 84000, Surat Thani, Thailand; jitralada.k@psu.ac.th (J.K.); kanokorn.s@psu.ac.th (K.S.-U.); charoen.n@psu.ac.th (C.N.); yutthapong.p@psu.ac.th (Y.P.)
2 Research and Development Center, Chalermkarnchana University, Mueang Sisaket 33000, Sisaket, Thailand; rattikalnearn@gmail.com
3 Department of Wood Industry, Faculty of Applied Sciences, Universiti Teknologi MARA Pahang Branch Campus Jengka, Bandar Tun Razak 26400, Pahang, Malaysia; leesenghua@uitm.edu.my
4 Faculty of Furniture Design and Wood Engineering, Transilvania University of Brasov, Universitatii 1, 500068 Brasov, Romania; emilia.salca@unitbv.ro
* Correspondence: aujchariya.c@psu.ac.th

**Abstract:** The purpose of this research was to investigate the production and properties of wood plastic pellets (WPP) made from rubberwood sawdust and refuse-derived fuel (RDF). WPP samples were tested for chemical and physical properties and compared to standard wood pellets. The results showed that when using RDF, the elemental compositions of WPP can affect the content of Zn, Cu, Pb, Cd, Cr, and As. In addition, RDF samples had a higher heating value of 21.19–22.09 MJ/kg. The physical properties of the samples revealed that they had a density of 1175–1286 kg/m$^3$, a mechanical durability of 98%, and a moisture content of 5.38–11.27%. According to the study's findings, these manufactured mixed pellets have the potential to be beneficial for alternative sustainable green energy as fuels. Moreover, using RDF, which comes from MSW, could help in global warming mitigation.

**Keywords:** wood plastic pellets; rubberwood sawdust; refuse-derived fuel; elemental compositions; chemical and physical properties

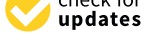



## 1. Introduction

Due to fossil fuel reserves being depleted, the world is shifting toward the production of renewable energy. Biomass materials are one of the most intriguing alternatives due to their widespread availability. Because of their low bulk density, biomass materials are difficult to handle, store, and transport. Pelletization can densify the materials into pellets by using pressure and heat [1]. The use of raw materials, such as agricultural, forestry, and food waste, and surpluses from the agri-food industry has recently increased dramatically [2]. Wood pellets are a cost-effective, environmentally friendly alternative to traditional fossil fuels. In terms of calorific value, softwoods had a higher range between 19.66 and 20.36 MJ/kg, while hardwoods ranged between 17.63 and 20.81 MJ/kg [3]. Sawdust, planer shavings, and dry chips have been the primary source materials used in the production of wood pellets in Sweden since 2001.

Because of their widespread availability, alternative forms of biomass, such as bark and logging residues, have sparked interest as raw materials [4]. A previous study [5] investigated the physical, mechanical, and energy properties of wood pellets derived from three common tropical species: *Acacia wrightii*, *Ebenopsis ebano*, and *Havardia pallens*. According to the findings, the pellets obtained from the species have a high energy density, making them ideal for commercial and industrial heating applications. Aside from wood residues and uncommon species, a variety of biomasses, including almond shells, olive stones, coffee dregs, coffee husks, grape pomace, hazelnut shells, miscanthus, pine kernel

shells, and switchgrass, were blended to produce high-quality pellets for industrial use [1]. Similarly, Carroll and Finnan [6] found that willow, miscanthus, wheat, barley, and rape straws were also used. Additionally, Limhengha et al. [7] showed the potential of empty fruit bunches as a biomass source for pellet production.

Thailand can grow plants all year, ensuring a steady supply of biomass feedstock. Thailand exported 172,441 tonnes of wood pellets to South Korea and Japan in 2019. Thai entrepreneurs want to increase wood pellet exports to Japan for power generation. According to the Kasikorn Research Center, Thailand will export 80,000–100,000 tonnes of wood pellets to Japan in 2020 [8]. The Thai government restricts the use of wood pellets to industrial heat production rather than electricity generation. In Thailand, biomass power plants use less expensive substitutes, such as palm shells, paddy, and bagasse, instead of wood pellets [9].

Rubberwood has the potential to produce a substantial amount of energy. It outperforms fruit fibers, rice husks, coconut husks, bagasse, and logging residues, and it even slightly outperforms wood residues. It does, however, have less potential than empty fruit bunches and palm shells. Rubberwood has lower ash and nitrogen content compared to oil palm biomass. There are numerous opportunities in Thailand for wood pellet production using para-rubber waste wood as a feedstock. Consumption of wood pellets can help reduce greenhouse gas emissions while also supporting the Thai government's efforts to use wood pellets for domestic purposes. Furthermore, there is a growing demand for wood pellets for export [10]. The global market for pellets is expanding. The output in 2008 was 9.8 $T_g$, with a global increase to 14.3 $T_g$ in 2010 and an excess of 26 $T_g$ in 2015 [11]. Because of the high demand for rubberwood in the wood product manufacturing industry, the quantity of rubberwood biomass available for energy production is relatively low compared to the abundance of oil palm biomass. Rubberwood biomass utilization as a fuel source is severely limited due to (1) systematic replanting, (2) extraction of all above-ground biomass up to 10 cm in diameter from the field, and (3) utilization of waste and residue from secondary milling activities in the panel products sector [12].

Pretreatment and blending diverse raw materials for co-pelletizing is a promising approach for improving biomass pellet quality, and co-pelletizing requires cost-effective, eco-friendly, and sustainable raw materials. Furthermore, in the coming years, co-pelletizing will be a viable option for increasing the competitiveness of large-scale biomass pellet fuel production [13]. Co-pelleting of woody and herbaceous crops; co-pelleting of crops containing high amounts of starch and oil, such as microalgae and peanut shells, with woody or herbaceous crops; mixed pelleting of different parts of the same crop, such as the bark and leaves of similar quality; and co-pelleting of biomass with solid waste, such as municipal sludge and paper waste, are examples of common co-pelleting combinations. The co-pelletization of non-biomass-based materials with biomass, such as household waste and inorganic additives, simplifies pelletizing and improves pellet fuel quality. The addition of 50% rubberwood sawdust and 50% lignite reduced the ash content by 50% [14].

Refuse-derived fuel (RDF) is a fuel substitute derived from waste management facilities. The use of RDF for heat and power production adheres to the European Union's waste hierarchy [15]. Materials such as refuse-derived fuel (RDF) fractions are used as fuel in cement or combined heat and power (CHP) plants. However, the low bulk density creates a number of transportation and storage issues [16]. In the case of biomass, these issues result in a decrease in pelletization. Because of its low bulk density, raw RDF has a high potential for densification operations. Furthermore, the negligible content of natural binders necessitates die redesign. Furthermore, studies have shown that temperature has a significant impact on pellet quality. For all variations of compaction pressure and channel diameter, the temperature condition of 120 °C produces the most durable pellets. This is due to plastics melting. Using appropriate operation combinations, RDF from municipal solid waste, packaging, wood, paper, and plastics could be more manageable and storable, with more predictable characteristics and specifications such as higher heating value and proximate and ultimate analyses of pellets [17]. RDF has gained value as an alternative fuel

due to the vast quantity of non-recyclable combustible waste materials, which can replace traditional generation burning fuels [18,19].

The impact of fossil fuels (coal, oil, and gas) on global warming is highly unsustainable for developing economies worldwide, and they are the most significant contributor to global climate change [20]. The use of biomass as a biofuel can communicate the need for low-carbon energy while also reducing pollution, paving the way to carbon neutrality. As a result, the critical decision of using municipal solid waste and biomass as sustainable energy resources to manage energy supply and demand issues, as well as climate change issues, must be made [21,22].

Previous literary works have revealed a potential ability to co-pelletize. To the best of the authors' knowledge, there have been few studies on the co-pelletization of rubberwood and RDF. As a result, there is a limited understanding of the effects of co-pelletization on the performance of the produced pellets. Our previous study produced mixed pellets from rubberwood and RDF and tested their mechanical durability and calorific value (ultimate and proximate) [23]. Meanwhile, the goal of this research is to determine the physical properties, higher heating value (HHV), and elemental analysis of pellets made from a rubberwood and RDF mixture. The findings of this research could support the viability of mixed pellet fuel as a feedstock in thermal waste-to-energy technologies.

## 2. Materials and Methods

### 2.1. Sample Preparation and Pelletization

RDF was selected and separated from a landfill area in Pattalung Province, Thailand, where the municipal solid waste (MSW) used in this study had been landfilled for 2 years and managed by the Provincial Administrative Organization of Pattalung Province. Firstly, MSW was recovered and separated into combustible and incombustible materials, using a separating machine and manual selection by workers. Then, the combustible materials were converted by a machine into refuse-derived fuel (RDF) level 3, which only has fluff RDF, such as plastics, papers, and fibers. Eventually, the RDF wastes were shredded to a size suitable for pelletization.

Rubberwood sawdust, a waste product supplied by BNS Wood Industry Co., Ltd., a sawmill located at Mueang, Surat Thani, Thailand, was collected. The material was controlled through its moisture content reaching 10–15% in a laboratory oven. The oversized sawdust was screened using a sieve with 18 meshes before mixing with RDF.

The composition of sawdust/RDF materials with four different types of pellets was prepared in ratios of 100/0, 70/30, 60/40, and 50/50 by weight, respectively. A rotary drum mixer was used to completely mix the two materials for 5 min before pelletization. The mixed materials were pelleted in an electric flat die wood pellet mill, ZhengZhou Known Imp. & Exp. Co., Ltd, KN-D-200, Zhengzhou, Henan, China, with 7.5 hp (380 v), 50 Hz, with the extruded temperature of 90–110 °C and a pellet mill die of 6 mm to produce 10 kg of each condition. Five samplings of each condition were randomly collected for the testing (Figure 1), while the pellets test had five replications of each property.

### 2.2. Physical Properties of the Samples

Dimensions such as the diameter and length of the pellets were measured with precisions of 0.01 mm. The density of samples was calculated by mass/volume, where pellets were weighed with precisions of 0.001 g. Mechanical durability testing of the pellets was carried out in accordance with the procedure outlined in EN 15210-1 [24]. The water absorption (WA) of the samples was evaluated with the water soaking method. The samples were soaked in distilled water in a 250 mL beaker with 100 mL of water for 5 min. After 24 h, the samples were taken out and drenched through filter paper—which had a pore size of 11 μm and was 125 mm in diameter—for 10 min [24]. The WA samples were measured by weight with precision of 0.01 g using the following equation:

$$\text{WA} = (M_{\text{after}} - M_{\text{before}})/M_{\text{before}} \times 100 \tag{1}$$

where WA is the water absorption of the pellet (%); $M_{after}$ is the mass of the pellet after soaking (g); $M_{before}$ is the mass of the pellet before soaking (g).

**Figure 1.** The completely randomized samples of each condition manufactured with sawdust/RDF ratios, (**a**) 100/0, (**b**) 50/50, (**c**) 60/40, and (**d**) 70/30.

*2.3. Microstructure Evaluation of the Samples*

A scanning electron microscope (SEM), FEI Quanta 250, Waltham, MA, USA, was used to image the samples. Before testing, the samples were coated with a thin layer of gold. The images were taken with a SEM set to high vacuum (HV) mode at 15 kV. Pellet cross-sections and longitudinal sections were photographed.

*2.4. Proximate Analysis of the Samples*

The proximate analysis of samples typically determined the moisture content (MC), volatile matter (VM), ash content (AC), and fixed carbon (FC) as a percentage. The percentage of fixed carbon was determined by using the following equation:

$$FC = 100 - (MC + VM + AC)\ \%\tag{2}$$

where FC is the fixed carbon (%); MC is the moisture content (%); VM is the volatile matter (%); and AC is the ash content (%).

*2.5. Higher Heating Value of the Samples*

The higher heating value (HHV) or gross calorific value (GCV) of pellet samples was determined by using an isoperibol bomb calorimeter, Leco AC500, St. Joseph, MI, USA. The measurements were taken in five replicates, and the results are given as means with standard deviations in MJ/kg following ASTM D 5865-13 [25]. The HHV of the samples as a function of fixed carbon (FC, wt%) was calculated using the following equation:

$$HHV = 0.196(FC) + 14.119\tag{3}$$

for which the correlation coefficient was 0.9997.

*2.6. Ultimate Analysis of the Samples*

The samples were analyzed using the ASTM D5373-93 (1997) [26] procedure to determine the contents of elemental carbon (C), hydrogen (H), nitrogen (N), and sulfur (S) using a Perkin Elmer, 2400 Series II CHNS/O analyzer, Waltham, MA, USA.

### 2.7. Elemental Analysis of Samples

The elemental analysis of samples finally analyzed their compositions. The potassium (K), sodium (Na), cadmium (Cd), zinc (Zn), copper (Cu), lead (Pb), and chromium (Cr) content was determined by using a high-performance flame AA spectrometer, PinAAcle 900F, Perkin Elmer, USA. The chlorine (Cl) content was determined by using a 785 DMP Titrino, Metrohm, Herisau, Switzerland. The arsenic (As) content was determined by using a FIAS 100, Perkin Elmer, USA. Ultimately, mercury (Hg) was determined using a FIMS 100, Perkin Elmer, USA.

### 2.8. FTIR-ATR Spectral Analysis of Samples

The FTIR spectroscopy of samples was performed using a Spectrum Two FT-IR Spectrometer (DTGS Detector), PerkinElmer, Llantrisant, UK, in order to compare the spectra of each pellet condition at 400–4000 cm$^{-1}$ resolution. The FTIR analysis was conducted quantitatively, with consideration given to the shape of the spectra at specific peaks and the intensity of the FTIR graph.

### 2.9. Data Analysis

A completely randomized design of sample types was used for this experiment. The analysis of variance (ANOVA) was used to determine the significant differences between the four types of wood pellet specimens, and Duncan's multiple range tests were used for additional analysis using SPSS Statistics version 22, IBM SPSS Statistics for Windows, Armonk, NY, USA. A *p*-value of 0.05 was used as the level of confidence.

## 3. Results and Discussion

### 3.1. Physical Characteristics of Pellets

The mean values of diameter, length, density, and mechanical durability of the samples are shown in Table 1. The diameter of the pellets did not differ much, ranging from 6.12 to 6.19 mm. The pellet with the addition of RDF was slightly bigger than the control sample (WPP). It seems that the size of the pellets increased with the increasing ratio of RDF. Additionally, the pellets with the addition of RDF were longer than those with WPP. However, WPP had a higher density than the RDF samples. The density of WPP was 1.29 g/cm$^3$, while the RDF samples had a density ranging from 1.12 to 1.24 g/cm$^3$. This experiment used mixing ratios by weight, and by adding RDF, the density decreased compared to the wood pellet; however, having a higher RDF ratio could increase the density of WPP. The highest density was observed in the sample with 50% RDF addition. The findings were in line with the study by Laosena et al. [23]. When the pellets had a higher RDF content, the water absorption (WA) of the samples was lower. Because plastic is a hydrophobic material, mixing RDF with the pellet can reduce WPP's moisture uptake. According to this study, having a 50/50 (Wood/RDF) ratio reduced the WA of the sample by 57%. The results showed that the WA values of WPP soaked in water for 5 min ranged from 4.17 to 6.83%, while wood pellets had a value of 9.68% [27].

**Table 1.** Physical characteristics of pellets.

| Sample Type (Sawdust/RDF) | Sample Type | Diameter (mm) Mean | | Length (mm) Mean | | Density (g/cm$^3$) Mean | | Water Absorption (%) Mean | | Mechanical Durability (%) Mean | |
|---|---|---|---|---|---|---|---|---|---|---|---|
| 100/0 | Control | 6.12 | ±(0.09) | 36.48 | ±(0.44) | 1.29 [d] | ±(0.08) | 9.68 [d] | ±(0.42) | 98.48 [a] | ±(0.30) |
| 70/30 | 70/30 | 6.18 | ±(0.01) | 42.81 | ±(0.77) | 1.12 [a] | ±(0.04) | 6.83 [c] | ±(0.59) | 98.26 [a] | ±(0.06) |
| 60/40 | 60/40 | 6.15 | ±(0.04) | 40.81 | ±(1.12) | 1.18 [b] | ±(0.15) | 5.56 [b] | ±(0.28) | 98.87 [b] | ±(0.04) |
| 50/50 | 50/50 | 6.19 | ±(0.03) | 41.29 | ±(1.35) | 1.24 [c] | ±(0.12) | 4.17 [a] | ±(0.50) | 99.06 [b] | ±(0.10) |

Numbers in parentheses are standard deviation values. Mean values with the different letters are significantly different at *p* < 0.05.

Meanwhile, the WPP had a mechanical durability of 98.48%. The results showed that the samples produced in this study are of high quality, with a mechanical durability value greater than 96%. However, the mechanical durability of the pellets decreased initially when 30% RDF was added but increased as the RDF ratio increased. The highest mechanical durability of 99.06% was recorded in the samples with 50% RDF addition. This phenomenon may be due to the strong association with the density of the pellets, as reported by Jewiarz et al. [16]. Pellets with 50% RDF addition had the highest density, and therefore, this may be the reason they showed the greatest durability among the tested samples.

### 3.2. SEM Images of Pellets

Figure 2 depicts the SEM images of rubberwood samples (Figure 2a,b) and the pellet made of wood and RDF at a 60/40 ratio (Figure 2c,d). The textures of the pellets differed from one another. The cross-section of the rubberwood pellet (Figure 2a) was coarse, revealing the wood fibers, whereas the longitudinal section (Figure 2b) was smooth. The pellet with RDF addition, on the other hand, showed the finished blending with smooth areas on the pellet's cross-section (Figure 2c) and longitudinal section (Figure 2d). The smooth surface indicates lower porosity, and thus, the water absorption of RDF pellets was lower than that of pellets made entirely of wood.

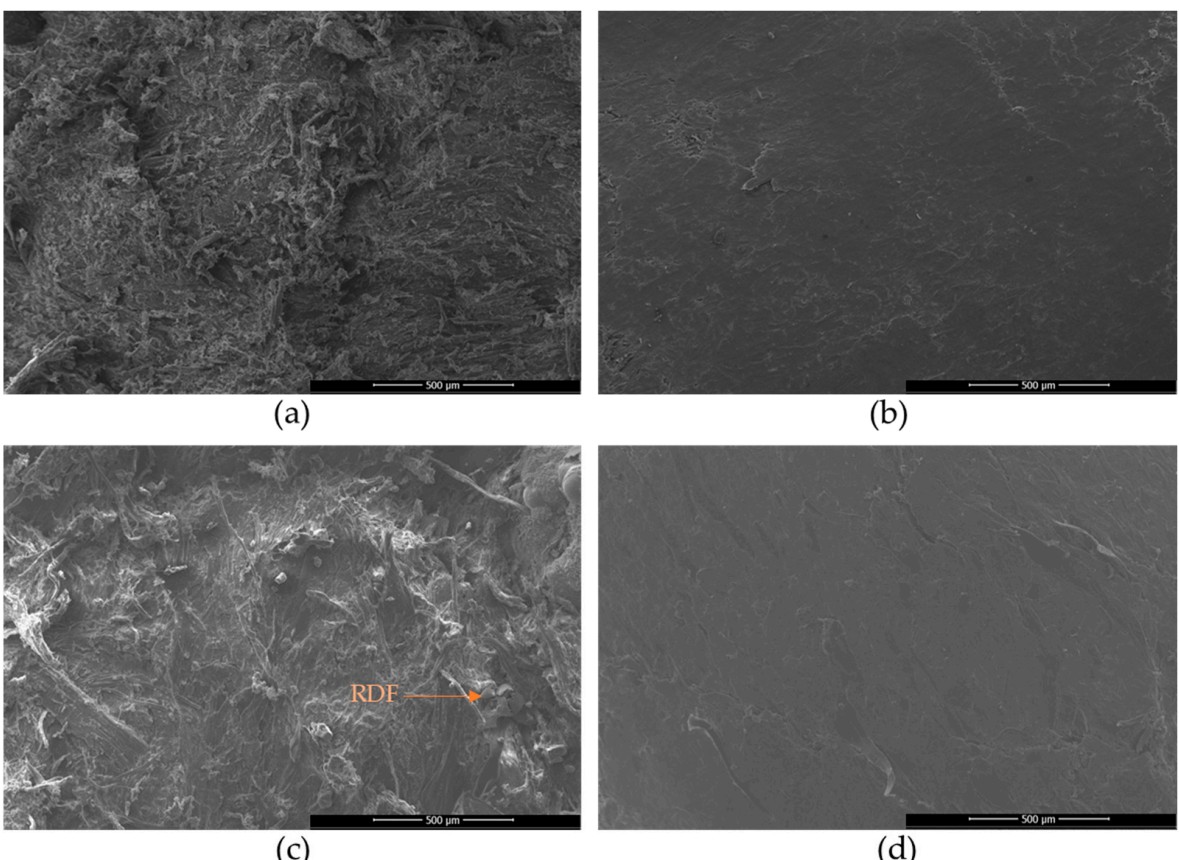

**Figure 2.** The SEM images of (**a**) cross-section of wood pellet, (**b**) longitudinal section of wood pellet, (**c**) cross-section of wood/RDF pellet, and (**d**) longitudinal section of wood/RDF pellet.

### 3.3. Proximate Analysis and Higher Heating Values (HHV) of Pellets

Table 2 shows the results of the experiment's proximity analysis and higher heating values (HHV). The moisture content of the samples ranged from 5.38 to 11.27%. Compared to the RDF samples (65.17 to 68.17%), WPP had the highest volatile matter of 72.11%. Meanwhile, the fixed carbon of pellets decreased when RDF was added. Furthermore, the ash content of the pellets was unsatisfactory in this experiment, with the control pellet



(2.54%) having less ash content (9.85–13.86%), which was 3.9–5.4 times lower than that of RDF samples. However, RDF reported an ash content of 11–13.4% [28]. The increase in the RDF ratio increased the volatile matter and ash content of the pellets while decreasing the fixed carbon content. Kramens et al. [28] conducted a similar study using pinewood and non-recyclable textile materials. On the other hand, the HHV of the pellets produced in this study ranged between 19.40 and 22.09 MJ/kg, whereas the HHV of typical RDF values were reported at 13–20 MJ/kg [29]. The results showed that adding RDF to the pellet significantly increased its HHV. The highest higher heating values of 22.09 MJ/kg were observed in the pellet with 50% RDF, while values of 19.40, 21.19, and 21.83 MJ/kg resulted in 100/0, 70/30, and 60/40 ratios, respectively. A previous investigation established that wood pellets made from mixed wood species had HVV of 19.85–20.45 MJ/kg [30], whereas an ultimate analysis of RDF in Europe reported an average HVV of 20.79 MJ/kg [31]. The highest HHV found in 50% RDF samples might be due to their lower MC and higher volatile matter and ash content. Ozkan et al. [32] demonstrated that the HHV of RDF pellets were inversely proportional to the MC but positively proportional to the volatile matter, ash content, and fixed carbon content.

**Table 2.** Proximate analysis and higher heating values of samples.

| Sample | Proximate Analysis | | | | | | | | HHV | |
|---|---|---|---|---|---|---|---|---|---|---|
| Type (Sawdust/RDF) | MC Mean (%) | | VM Mean (%) | | AC Mean (%) | | FC Mean (%) | | (MJ/kg) Mean | |
| 100/0 | 9.57 [b] | ±(0.26) | 72.11 [d] | ±(0.31) | 2.54 [a] | ±(0.09) | 15.78 [b] | ±(0.56) | 19.40 [a] | ±(0.07) |
| 70/30 | 11.27 [d] | ±(0.31) | 65.87 [b] | ±(0.36) | 9.85 [b] | ±(0.11) | 13.01 [a] | ±(0.45) | 21.19 [b] | ±(0.09) |
| 60/40 | 10.48 [c] | ±(0.06) | 65.17 [a] | ±(0.36) | 11.86 [c] | ±(0.15) | 12.48 [a] | ±(0.40) | 21.83 [c] | ±(0.14) |
| 50/50 | 5.38 [a] | ±(0.35) | 68.17 [c] | ±(0.49) | 13.86 [d] | ±(0.16) | 12.59 [a] | ±(0.55) | 22.09 [d] | ±(0.08) |

Numbers in parentheses are standard deviation values. Mean values with the different letters are significantly different at $p < 0.05$.

### 3.4. Ultimate Analysis Values of Pellets

Table 3 shows the percentages of carbon (C), hydrogen (H), nitrogen (N), and sulphur (S) in the control and RDF samples tested. The control samples had C, H, N, and S content of 51.20%, 8.01%, 0.27%, and 0.60%, respectively. In the case of RDF samples, the C content ranged from 47.46 to 49.87%. When RDF was added, the C content decreased, but it increased as the RDF ratio increased. On the contrary, the H, N, and S contents of the RDF samples were greater than those of control samples. Generally, the C content of the RDF samples produced in this study was higher than the RDF pellets produced by Kobyashi et al. (42–44%) [33] but comparable to those produced in Italy (49%) [34].

**Table 3.** Ultimate analysis values of samples.

| Sample | Sample Type | Ultimate Analysis (%) | | | | | | | |
|---|---|---|---|---|---|---|---|---|---|
| Type (Sawdust/RDF) | | C Mean | | H Mean | | N Mean | | S Mean | |
| 100/0 | Control | 51.20 [a] | ±(7.24) | 8.01 [a] | ±(1.13) | 0.27 [a] | ±(0.03) | 0.60 [a] | ±(0.03) |
| 70/30 | 70/30 | 47.46 [a] | ±(0.27) | 10.21 [b] | ±(1.10) | 0.47 [b] | ±(0.03) | 0.91 [b] | ±(0.02) |
| 60/40 | 60/40 | 47.67 [a] | ±(0.34) | 11.45 [b] | ±(0.69) | 0.50 [b] | ±(0.13) | 1.38 [c] | ±(0.15) |
| 50/50 | 50/50 | 49.87 [a] | ±(1.58) | 10.38 [b] | ±(1.00) | 0.69 [c] | ±(0.04) | 1.04 [b] | ±(0.08) |

Numbers in parentheses are standard deviation values. Mean values with the different letters are significantly different at $p < 0.05$.

### 3.5. Elemental Analysis Values of Pellets

Table 4 reports the ten elemental compositions of the pellets. According to Table 4, there was no significant difference in the K and Hg content between WPP samples and RDF samples. Meanwhile, the remaining elemental compositions varied greatly. For instance,

RDF samples had significantly higher Na, Cl, Zn, Cu, Pb, Cd, Cr, and As content compared to that of WPP samples. Additionally, ANOVA testing showed a significant difference between groups and within groups of the sample type, as displayed in Table 5. Generally, pellets with the highest RDF ratio (50%) had the highest value of all elemental compositions. RDF is a fuel made from various wastes, including municipal solid waste (MSW) and industrial waste. MSW is primarily composed of various types of waste, such as plastics, textiles, rubber, and foam, which contain varying amounts of elemental components [35]. As a result, it is understandable that the RDF samples had more elemental components than the WPP samples.

**Table 4.** Elemental analysis values of samples.

| Sample Type (Sawdust/RDF) | Elemental Analysis | | | | | | | | | |
|---|---|---|---|---|---|---|---|---|---|---|
| | K Mean (%) | Na Mean (%) | Cl Mean (%) | Zn Mean (mg/kg) | Cu Mean (mg/kg) | Pb Mean (mg/kg) | Cd Mean (mg/kg) | Cr Mean (mg/kg) | As Mean (mg/kg) | Hg Mean (mg/kg) |
| 100/0 | 0.228 [a] ±(0.01) | 0.016 [a] ±(0.001) | 0.017 [a] ±(0.001) | 10.190 [a] ±(1.78) | 5.475 [a] ±(0.94) | 27.163 [a] ±(3.05) | 0.001 [a] ±(0.00) | 19.014 [a] ±(1.79) | 0.180 [a] ±(0.01) | 0.0001 [a] ±(0.00) |
| 70/30 | 0.237 [a] ±(0.09) | 0.101 [b] ±(0.003) | 0.092 [b] ±(0.004) | 328.492 [b] ±(16.90) | 61.763 [b] ±(5.67) | 57.449 [bc] ±(7.82) | 1.517 [b] ±(0.20) | 44.014 [b] ±(2.22) | 2.306 [b] ±(0.16) | 0.0001 [a] ±(0.00) |
| 60/40 | 0.234 [a] ±(0.01) | 0.122 [c] ±(0.002) | 0.116 [c] ±(0.003) | 380.599 [c] ±(7.74) | 70.076 [b] ±(11.09) | 48.275 [b] ±(7.51) | 2.287 [c] ±(0.14) | 64.274 [c] ±(3.45) | 2.927 [c] ±(0.19) | 0.0001 [a] ±(0.00) |
| 50/50 | 0.237 [a] ±(0.01) | 0.144 [d] ±(0.004) | 0.125 [d] ±(0.002) | 400.015 [d] ±(12.27) | 70.496 [b] ±(3.43) | 61.996 [c] ±(5.34) | 3.360 [d] ±(0.34) | 64.735 [c] ±(5.34) | 2.725 [c] ±(0.34) | 0.0001 [a] ±(0.00) |

Numbers in parentheses are standard deviation values. Mean values with the different letters are significantly different at $p < 0.05$.

**Table 5.** The ANOVA testing of some elemental compositions of samples.

| | | Sum of Squares | df | Mean Square | F | Sig. |
|---|---|---|---|---|---|---|
| Zn | Between Groups | 498,363.018 | 3 | 166,121.006 | 1063.927 | 0.000 |
| | Within Groups | 2498.231 | 16 | 156.139 | | |
| | Total | 500,861.250 | 19 | | | |
| Cu | Between Groups | 14,643.652 | 3 | 4881.217 | 92.952 | 0.000 |
| | Within Groups | 840.214 | 16 | 52.513 | | |
| | Total | 15,483.865 | 19 | | | |
| Pb | Between Groups | 3586.758 | 3 | 1195.586 | 24.576 | 0.000 |
| | Within Groups | 778.374 | 16 | 48.648 | | |
| | Total | 4365.132 | 19 | | | |
| Cd | Between Groups | 29.941 | 3 | 9.980 | 177.785 | 0.000 |
| | Within Groups | 0.898 | 16 | 0.056 | | |
| | Total | 30.839 | 19 | | | |
| Cr | Between Groups | 7004.878 | 3 | 2334.959 | 253.536 | 0.000 |
| | Within Groups | 147.353 | 16 | 9.210 | | |
| | Total | 7152.231 | 19 | | | |
| As | Between Groups | 23.944 | 3 | 7.981 | 112.429 | 0.000 |
| | Within Groups | 1.136 | 16 | 0.071 | | |
| | Total | 25.080 | 19 | | | |

*3.6. FTIR Spectra of Pellets*

Figure 3 shows the changes in function groups of different wood/RDF pellet samples. In 100/0 pellet samples with 100% wood composition, the broad absorption near the peaks around 3300–3400 cm$^{-1}$ and 2800–3000 cm$^{-1}$ corresponds to the hydrogen-bonded O-H stretch and C-H stretch, respectively. These structures are commonly found in lignocel-

lulosic material [36], which is rubberwood in this case. After mixing with RDF, two new sharp peaks at around 2800–3000 cm$^{-1}$ can be seen. These peaks could correspond to the C-CH$_3$ groups [37], which may belong to the plastic waste in the RDF. The peaks, including those around 3000–3400 cm$^{-1}$, become less intense as the proportion of RDF increases, most likely due to the successful cross-linking during rubberwood and RDF co-pelletization, especially when RDF is added at a higher proportion.

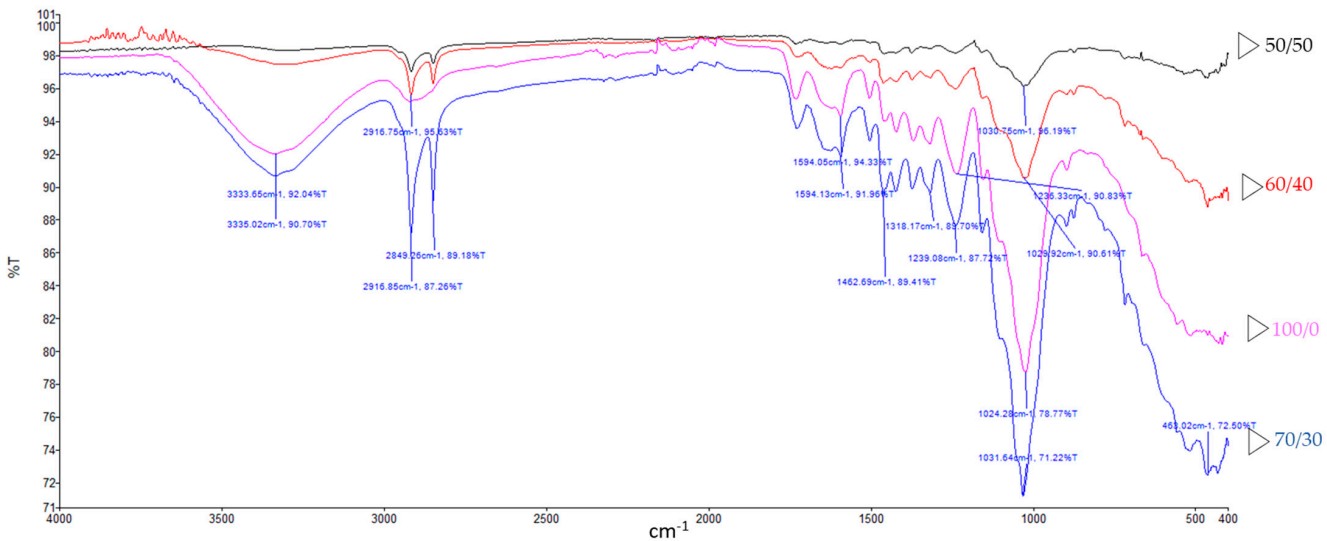

**Figure 3.** The FTIR spectra of samples with the 100/0, 70/30, 60/40, and 50/50 ratios of wood/RDF.

## 4. Conclusions

This study provided a comprehensive foundation for wood plastic pellets made from RDF and rubberwood sawdust. Specimen elemental compositions were compared with the standard. Furthermore, the information on their properties revealed a significant difference between each component ratio at a 95% confidence level. The HHV of RDF samples increased significantly, while the ash content needs to be reduced. However, using RDF derived from MSW as an alternative sustainable green energy fuel could aid in global warming mitigation. Therefore, it can be concluded that wood plastic pellets have the potential to be used as a biofuel in a clean industry. Moreover, this product may also be useful for composite materials, such as building materials, which previous studies had examined in order to promote plastic waste use as a substitute for concrete components, which improve the mechanical properties of the materials [38,39]. In a further study, the properties of light weight concrete made from WPP and Portland cement will be investigated in order to attain green materials.

**Author Contributions:** Conceptualization, A.C. and R.L.; Data curation, A.C.; Formal analysis, R.L., K.S.-U. and J.K.; Investigation, R.L., J.K. and K.S.-U.; Methodology, A.C.; Project administration, A.C.; Resources, J.K. and K.S.-U.; Supervision, E.-A.S., S.H.L., Y.P. and C.N.; Writing—Original draft, K.S.-U., J.K. and A.C.; Writing—Review and editing, C.N., S.H.L., E.-A.S. and A.C. All authors have read and agreed to the published version of the manuscript.

**Funding:** This project was funded by a National Research Council of Thailand (NRCT) grant allocated to Prince of Songkla University, Grant Number N71A660396, and the APC was also funded. This research was also funded by a grant from the Interdisciplinary Graduate School of Energy Systems (IGS-Energy), Prince of Songkla University, allocated to Rattikal Laosena.

**Institutional Review Board Statement:** Not applicable.

**Informed Consent Statement:** Not applicable.

**Conflicts of Interest:** The authors declare that there are no conflict of interest regarding the publication of this paper.

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
