# Peer review of "Elemental Compositions of Wood Plastic Pellets Made from Sawdust and Refuse-Derived Fuel (RDF) Waste"

_applsci, doi:10.3390/app132011162_

Round 1
Reviewer 1 Report
1. In the introduction, the author emphasizes that wood pellets are used as fuel, so why does this reduce greenhouse gas emissions?
2. why does the density of WPP decrease and then increase as the RDF content increases?
3. what is the purpose of determining the mechanical durability of WPP when it is used as a fuel? In Table 1, the mechanical durability of all the samples is not much different from each other, so it is recommended to do a significance analysis.
none
Author Response
1.Comments and Suggestions for Authors
- In the introduction, the author emphasizes that wood pellets are used as fuel, so why does this reduce greenhouse gas emissions?
- The information was added in the introduction to show the advantage of using biomass to reduce GHG.
2. Why does the density of WPP decrease and then increase as the RDF content increases?
- The information was described line 211.
3. What is the purpose of determining the mechanical durability of WPP when it is used as a fuel? In Table 1, the mechanical durability of all the samples is not much different from each other, so it is recommended to do a significance analysis.
-The mechanical durability of pellets needs to examine because the transportation can damage them to powder and the significance analysis of this experiment was added.

Reviewer 2 Report
In the manuscript (applsci-2634877), entitled “Elemental compositions of wood plastic pellets made from sawdust and refuse-derived fuel (RDF) waste” the production and properties of wood plastic pellets (WPP) made from rubber wood sawdust and refuse-derived fuel (RDF) was investigated. There are many studies in the field of study so this manuscript seems an ordinary study. In addition, the quality of the manuscript should be increased. Related points are summarized below.
1) What is the pelletization temperature in section 2.1.
2) Instead of presenting a box of the pellets a closer image of a single of them can be presented in Figure 1.
3) Please revise the following sentence
Line 207: “ The highest HHV of 22.09 MJ/kg …” the highest higher heating values !!
4) a, b, c, and d indices in the tables should be defined in each subtitle.
5) Please present .000 as 0.000 in Figure 5.
6) What is the main improvement? The performance of the material should be tested with the other published studies.
7) The results should also be sported with the chemical analysis methods ie FTIR, SEM etc.
Author Response
2.Comments and Suggestions for Authors
In the manuscript (applsci-2634877), entitled “Elemental compositions of wood plastic pellets made from sawdust and refuse-derived fuel (RDF) waste” the production and properties of wood plastic pellets (WPP) made from rubber wood sawdust and refuse-derived fuel (RDF) was investigated. There are many studies in the field of study so this manuscript seems an ordinary study. In addition, the quality of the manuscript should be increased. Related points are summarized below.
1) What is the pelletization temperature in section 2.1.
- The information was added in line 140.
2) Instead of presenting a box of the pellets a closer image of a single of them can be presented in Figure 1.
- Figure 1 was revised.
3) Please revise the following sentence
Line 207: “ The highest HHV of 22.09 MJ/kg …” the highest higher heating values !!
- It was revised.
4) a, b, c, and d indices in the tables should be defined in each subtitle.
- The definition was added.
5) Please present .000 as 0.000 in Figure 5.
- It was revised.
6) What is the main improvement? The performance of the material should be tested with the other published studies.
- The information and more citations were added.
7) The results should also be sported with the chemical analysis methods ie FTIR, SEM etc.
- FTIR and SEM results were illustrated in a revised manuscript.

Reviewer 3 Report
Paper entitled "Elemental compositions of wood plastic pellets made from sawdust and refuse-derived fuel (RDF) waste" of authors Aujchariya Chotikhun, Rattikal Laosena, Jitralada Kittijaruwattana, Seng Hua Lee, Kanokorn Sae-Ueng, Charoen Nakason, Yutthapong Pianroj, Emilia-Adela Salca is a very interesting paper.
In this paper the authors presented investigate the production and properties of wood plastic pellets (WPP) made from rubberwood sawdust and refuse-derived fuel (RDF)
Authors presented different analysis methods in order compared to standard wood pellets. The results showed that when using RDF, the elemental compositions of WPP can affect the content of Zn, Cu, Pb, Cd, Cr, and As. In addition, RDF samples had a higher heating value of 21.19 - 22.09 MJ/kg. The physical properties of the samples revealed that they had a density of 1,175 - 1,286 kg/m3, a mechanical durability of 98%, and a moisture content of 5.38 - 11.27%. According to the authors findings, these manufactured mixed pellets have the potential to be beneficial for alternative sustainable green energy as fuels.
However, this paper lacks experiments to confirm the specific application of the examined wood plastic pellets (WPP). Additionally, this paper lacks experiments to confirm the safety of use of the rubberwood sawdust and refuse-derived fuel (RDF) for the presence of heavy metals (for example ICP or AAS), as well as leaching and evaporation tests of heavy metals during and after application.
Moreover, the authors state that these products may also be useful for composite materials such as building materials, but there are no results confirming this.
I ask the authors to add it.
The paper is too short, with few results and references presented.
I recommend reconsider after major revision (control missing in some experiments).
Author Response
3.Comments and Suggestions for Authors
- Thank you for your comments and suggestions. I revised the manuscript that adding FTIR, SEM, and Water absorption testing. The information and more citations were added.

Round 2
Reviewer 2 Report
The required changes have been made and it can be accepted after a minor version. Related points are presented below.
1) Which pellet was used in Figure 2.
2) Please use only Figure or Fig abbreviation in the whole manuscript including text.
3) Line 258: Please revise the sentence " The highest higher heating values"
4) Please compare the performance of the pellet with the other published studies.
Author Response
1.Comments and Suggestions for Authors
The required changes have been made and it can be accepted after a minor version. Related points are presented below.
1) Which pellet was used in Figure 2.
- The information was added in line 234.
2) Please use only Figure or Fig abbreviation in the whole manuscript including text.
- All fig abbreviations were changed to Figure.
3) Line 258: Please revise the sentence " The highest higher heating values"
- The sentence was revised.
4) Please compare the performance of the pellet with the other published studies.
- The information and more citations were added.
Reviewer 3 Report
Paper entitled "Elemental compositions of wood plastic pellets made from sawdust and refuse-derived fuel (RDF) waste" of authors Aujchariya Chotikhun, Rattikal Laosena, Jitralada Kittijaruwattana, Seng Hua Lee, Kanokorn Sae-Ueng, Charoen Nakason, Yutthapong Pianroj, Emilia-Adela Salca is a very interesting paper.
In this paper the authors presented investigate the production and properties of wood plastic pellets (WPP) made from rubberwood sawdust and refuse-derived fuel (RDF).
The authors supplemented the work with different analysis methods, however, this paper still lacks experiments to confirm the specific application of the examined wood plastic pellets (WPP). Additionally, this paper still lacks experiments to confirm the safety of use of the rubberwood sawdust and refuse-derived fuel (RDF) for the presence of heavy metals (for example ICP or AAS), as well as leaching and evaporation tests of heavy metals during and after application.
Moreover, the authors state that these products may also be useful for composite materials such as building materials, but there are no results confirming this.
I ask the authors to add it.
I recommend reconsidering after minor revision (corrections to minor methodological errors and text editing).
Author Response
2.Comments and Suggestions for Authors
Paper entitled "Elemental compositions of wood plastic pellets made from sawdust and refuse-derived fuel (RDF) waste" of authors Aujchariya Chotikhun, Rattikal Laosena, Jitralada Kittijaruwattana, Seng Hua Lee, Kanokorn Sae-Ueng, Charoen Nakason, Yutthapong Pianroj, Emilia-Adela Salca is a very interesting paper.
In this paper the authors presented investigate the production and properties of wood plastic pellets (WPP) made from rubberwood sawdust and refuse-derived fuel (RDF).
The authors supplemented the work with different analysis methods, however, this paper still lacks experiments to confirm the specific application of the examined wood plastic pellets (WPP). Additionally, this paper still lacks experiments to confirm the safety of use of the rubberwood sawdust and refuse-derived fuel (RDF) for the presence of heavy metals (for example ICP or AAS), as well as leaching and evaporation tests of heavy metals during and after application.
- The author added the definition of RDF level 3 in line 130 that RDF has been processed to separate glass, metal, and inorganic materials by the machine. Therefore, the RDF 3 in this experiment has only fluff RDF such as plastics, papers, and fibers. However, I can provide the data of our experiment on ICP testing as a figure below that it is from a mixed pellet that will present in next manuscript.
Moreover, the authors state that these products may also be useful for composite materials such as building materials, but there are no results confirming this.
- The information and more citations were added.
